# A Comparison of Devices for Race Day Characterization of North American Turfgrass Thoroughbred Racing Surfaces

**DOI:** 10.3390/ani14010038

**Published:** 2023-12-21

**Authors:** Peter R. Schmitt, Wayne Sanderson, John (Trey) Rogers, Tyler J. Barzee, Michael (Mick) Peterson

**Affiliations:** 1Biosystems and Agricultural Engineering, University of Kentucky, Lexington, KY 40503, USAtjbarzee@uky.edu (T.J.B.); 2Racing Surfaces Testing Laboratory, Lexington, KY 40502, USA; 3Plant, Soil, and Microbial Sciences, Michigan State University, East Lansing, MI 48824, USA; rogersj@msu.edu

**Keywords:** equine, jockey, safety, turfgrass, racing, portable tools

## Abstract

**Simple Summary:**

The purpose of this study is to investigate the potential of using simple, portable tools to conduct surface condition measurements on race days at turfgrass Thoroughbred racetracks to approximate the biomechanical measurements made prior to the race meet. Test plot measurements were conducted with these tools as well as a more complex biomechanical device which replicates the speeds and loads of a Thoroughbred at a gallop. The turf plots were chosen to simulate a wide range of potential turf profiles that could be used in North American racetracks. The correlations were investigated and linear regression models were constructed to determine the level of approximation the simpler tools might achieve with the range of profiles considered. The volumetric moisture content was found to be the primary simple daily measurement which correlates to biomechanically based measurements. The penetration from the Longchamp Penetrometer and surface hardness from the Clegg Impact Hammer can further improve the approximation to equine biomechanics if desired. As this data are collected on a larger scale, it can then be paired with race times and injuries to investigate potential links between these measurements and horse performance and risk of injury.

**Abstract:**

Both pre-race meet and daily turf surface condition measurements are required by regulations adopted as part of the Horseracing Integrity and Safety Act (HISA). The Orono Biomechanical Surface Tester (OBST) is the primary device used for characterizing a racing surface and is used for the pre-meet inspections. Tools that are better suited for the daily testing of turf surfaces are also needed to meet the new federal regulations. The purpose of this study was to compare five simple tools commonly used in turf applications to the OBST. Data were collected with each of the six devices at plots chosen to approximate the current and potential compositions of North American turf racetracks. Correlations and linear regression models were then established between the simple tool measurements and the parameters measured by the OBST. The moisture probe was found to be the primary device for race day characterization due to its strong correlation to OBST measurements. The Longchamp Penetrometer is also prioritized for daily measurements due to its established correlation to horse performance and injuries. The Clegg Impact Hammer provides further improvement of the linear regression model. The Turf Shear Tester and GoingStick^®^ were not found to correlate well to the biomechanically based device.

## 1. Introduction

Thoroughbred racing has been a popular sport in North America since the American Revolution [1]. A more modern perspective on animal welfare and the safety of the riders has challenged the sport’s social license to operate [2]. Of particular concern are catastrophic injuries, which The Jockey Club defines as the death or euthanasia of the horse within 72 h of a race [3]. While the overall animal welfare is a concern, catastrophic injuries are a particular threat to the sustainability of horseracing due to a direct connection to the racing event. Research has also shown jockeys are at a significantly higher risk of injury when they are on a horse which experiences a catastrophic injury during a race [4].

While catastrophic injuries can result from a number of different sources, the condition of the racing surface is generally accepted as a particular concern since it is one of very few factors that affects all horses in a race [5]. An adequate turfgrass horse racing surface should allow the hoof to penetrate the surface for the purpose of providing stability of the center of mass [6] and reducing secondary impact loads [7] as well as providing adequate traction for the athlete for both straight line movement and turning [8]. These conditions should be met while experiencing the high loads and loading rates applied by a Thoroughbred racehorse at a full gallop [9]. A surface which does not allow for hoof penetration may not provide adequate grip and may increase the risk of high ground reaction forces and the associated risk of musculoskeletal injury [10]. Damage to a surface from divoting will result in an uneven surface for the horses following in a race or in later races. This can introduce loading moments in the mediolateral and craniocaudal axes which may be similar to previously established risk factors for lameness or even musculoskeletal disease [11].

The condition of the turf racing surface has been shown to impact the likelihood of injury to the horse [12,13] and jockey [14]. This evidence has resulted in regulations from the Horseracing Integrity and Safety Act (HISA) that has established requirements for surface condition measurements prior to the race meet and on race days [15]. Pre-race meet inspections for surfaces include the measurement of the mechanical properties with a surface tester based on the biomechanics of a Thoroughbred horse at a gallop [16]. While the biomechanically based measurements are required for pre-meet inspection [15], smaller and simpler devices are better suited for daily tests. The daily measurements prescribed by the HISA are moisture content as well as penetration and shear properties. This opens the possibility that standard turf testing tools can be used or tools that have been adopted in other countries to characterize turf racing surfaces. The test requirements help ensure the racing surface is as consistent as possible using pre-meet testing protocols as well as simpler tools to detect changes, such as moisture content, that occur over a shorter time period and impact risk [13]. These measurements are reasonably well-established for dirt surfaces, the most common Thoroughbred surface in North America. Turf, however, is the dominant surface in much of the world and has been gaining in popularity in North America [3]. Finding the most appropriate tools for measurements on turf differs by country [17,18,19] and only the United States has a biomechanically based system in general use [15].

The Orono Biomechanical Surface Tester (OBST) [16] is the primary method for evaluating an equine surface and is included as an international standard for the in situ testing of the functional properties of equine surfaces [20]. This device mimics the forelimb of a Thoroughbred at a gallop and is ideal for evaluating a racing surface prior to the race meet. The OBST’s potential use in daily data collection is limited due to the size and complexity of the test apparatus, but the direction set forth in ASTM F3400-19 should be adopted for daily measurements to the greatest extent possible. The functional parameters of cushioning, impact firmness, grip, and responsiveness are of particular importance.

There are a number of smaller tools which are more cost-effective than the OBST and cause less disruption to the racing surface, which would be beneficial for use on turf surfaces being actively used for racing. This study considers five readily available portable devices: a moisture probe, the Clegg Impact Hammer (CIH), the Longchamp Penetrometer (LP), the Turf Shear Tester (TST), and the GoingStick^®^. These devices have been used in similar applications (equine sports outside of North America, human sports, and turfgrass research), some of which have even been adopted as ASTM standards [21,22]. The five simple tools are compared to the OBST in an effort to establish connections to the functional parameters as defined in ASTM F3400-19.

The potential suitability of these tools must begin by investigating the correlations to the measurements taken at the speeds and loads of a Thoroughbred at a full gallop since these surfaces are non-linear and strain rate-dependent [23]. The measurements must also be applicable to the wide range of surface compositions used both currently and in future turfgrass racing surfaces [24], as prior research has shown the surface composition to affect characteristics such as surface hardness and divot resistance [25].

The intent of this study is to identify simple tools which are suitable for daily use and can provide quantitative data to describe the condition of Thoroughbred turfgrass racing surfaces on race days. Doing so would allow for racetracks to efficiently use their limited resources to obtain high-quality, repeatable, and objective data to assess the racing surface. The widespread use of standard methods for data collection on race days would complement pre-meet inspections [15] and also allow for future research to examine the potential correlations between measurements to both the performance and risk to the horse and rider.

## 2. Materials and Methods

### 2.1. Materials Tested

The test configurations included Kentucky Bluegrass and Bermudagrass species as well as several types of fiber reinforcements. Data were collected at Michigan State University’s Hancock Turfgrass Research Center in East Lansing, MI. The 15 different soil preparations were arranged in a randomized complete block design with three replications per treatment. Each plot measures 3.1 m by 4.9 m and has its rootzone separated from adjacent plots by below-ground walls constructed of oriented strand board. The design of the plots was carried over from prior studies [25,26], with only seasonal maintenance conducted. These plots were chosen as the subject of this study because they reflect some of the current as well as potential new compositions of North American turf tracks [24]. The treatments are described in Table 1, below.

### 2.2. Surface Condition Measurement Tools Used

The OBST replicates the motion from the point when the leading forelimb contacts the surface and the weight of the horse is transferred to the hoof. This is the point where the highest vertical and shear loads are applied. A total of four different parameters are calculated from the OBST [20,27]. The four parameters include cushioning (peak vertical load [kN]), impact firmness (peak vertical deceleration [g]), grip (fore/aft slide distance of the hoof [mm]), and responsiveness (time between peak spring compression velocity and maximum spring compression divided by the time between maximum spring compression and peak spring recoil velocity [%]). To collect data with the OBST, a 29.9 kg sled equipped with a hoof shaped projectile and a size 2 aluminum racing plate was dropped from a 1.43 m height at an angle of 8° from vertical. Data were collected by a tri-axial load cell, tri-axial accelerometer, a string potentiometer, and a linear potentiometer at a sampling rate of 10 kHz.

A FieldScout TDR 350 (Spectrum Technologies, Inc., Aurora, IL, USA) equipped with two 7.5 cm rods was used to measure volumetric moisture content (VMC [%]). These measurements are also included in ASTM F3400-19 due to VMC having long been shown to have an effect on the animal’s response to the condition of the racing surface [28]. Furthermore, VMC measurements can be conducted rapidly and repeatedly with very minimal effort using many different devices found on the open market. This is a distinct advantage for racetrack personnel who would be collecting data.

Surface hardness was measured with a Clegg Impact Hammer (CIH) (Lafayette Instrument Company, Lafayette, IN, USA). This device consists of a 2.25 kg mass with an integrated accelerometer which is dropped through a vertical guide tube from a height of 0.45 m onto the surface. Each time the mass is dropped onto the surface, the device provides a maximum acceleration value in CIT’s, which is then multiplied by 10 to obtain units of grams. While the CIH has not been widely adopted for the evaluation of equine surfaces, it has been used in turfgrass research and has a number of associated international standards for athletic field applications and construction [21,22].

A Longchamp Penetrometer (LP) was also used in this study to measure penetration. This device consists of a 1 kg mass which falls from 1 m onto a rod with a 1 cm^2^ surface area that penetrates the surface. Measurements for this study were conducted manually using the scale on the device and recorded in cm. Though there is no international standard for this device, it is commonly used in horse racing as it was developed for France Galop’s Longchamp Racecourse and it is used on a daily basis in a number of racing jurisdictions. What makes the Longchamp penetrometer unique among the other tools in this study is published research which correlates these measurements to both race times [29,30] and injuries [18] on turf racing surfaces. Unlike the track condition ratings used in most other racing jurisdictions, the penetrometer has been used to directly produce track ratings. Track conditions were “firm” for a penetrometer reading of 1.0–2.0, “good” for a penetrometer reading of 3.0–4.0, “soft” for a penetrometer reading of 5.0–7.0 and “heavy” for a penetrometer reading of 8–10 [18].

The Turf Shear Tester (TST) (Dr Baden Clegg Pty Ltd., Jolimont, Australia) was used to measure divot resistance. This device was fitted with a 50 mm wide shearing plate fixed at a depth of 40 mm. The device provides values of kgf on the display which were manually recorded and then multiplied by 9.8 to convert to N and then by a moment arm distance of 0.207 m to obtain units of N-m. While the TST has not yet been used in equine applications, it has been used in turfgrass research for evaluating the shear strength of those surfaces [25,31].

The GoingStick^®^ (Turftrax, Ltd., Cambridgeshire, UK) is a device that was developed through a collaboration between Turftrax, Ltd. and Cranfield University [19]. This device is used in the horse racing industry to quantify the “going” of turf racing surfaces in some jurisdictions although it is not associated with an industry standard. To collect data with the GoingStick^®^, the probe of the device is inserted into the ground vertically in a controlled manner (which produces a value for penetration) and the handle was then pulled back to produce a 45-degree angle with the surface (which produces a value for shear). The penetration and shear values are reported on the GoingStick’s unitless scale from 1–15, with 1 representing softer ground and 15 representing firmer ground [19]. These values can then be converted to SI units of N for penetration and N-m for shear [32]. The two values are combined into a composite “Going Index,” which is the value commonly used in the industry to characterize the racing surface. The Going Index was also calculated using a standard formula [19].

### 2.3. Testing Methods

The turfgrass plots did not receive any irrigation apart from natural rainfall for two weeks prior to testing to achieve a low moisture content. Data from a weather station located at the Hancock Turfgrass Research Center are included in Appendix A (Table A1). The experiment began by collecting data with the six different measurement tools. The plots were then irrigated for 1 h and data collection was repeated with the six tools. A total of 4 h of irrigation was then applied overnight. The data collection method was then repeated a third time, resulting in observations with each of the six tools at three different moisture levels.

One OBST drop was conducted at three random locations in each plot during each of the three data collection events. Data were then exported and post-processed in MATLAB to calculate the functional parameters of cushioning, impact firmness, responsiveness, and grip in accordance with the ASTM standard [20].

VMC was measured with the TDR 350 in accordance with ASTM D6780 [33]. Three measurements were taken per plot in random locations for each of the three data collection events. Three consecutive drops of the CIH were made without moving the guide tube and the deceleration value of each drop was recorded manually. This allows results to be reported in accordance with ASTM F1702-10 (which reports the first drop only) as well as based on ASTM F1936-07 (which reports the average of the second and third drops). These values are labeled CIH_1_ and CIH_23_, respectively, in this study. Though ASTM F1936-07 specifies the use of a 9.1 kg CIH, the 2.25 kg CIH is used in this study, which is consistent with ASTM F1702-10. Three measurement locations, with three drops each, were taken per plot in random locations in each of the three data collection events.

For each data collection event with the LP, a scale is read prior to and after releasing the 1 kg mass, with both values recorded manually. The reason for taking a penetrometer reading prior to dropping the mass is to account for potential surface irregularities such as minor elevation changes and the thatch layer of the turfgrass. Doing so allows for potential correlations to the OBST parameters to be investigated for the maximum penetration value as well as the difference between the maximum penetration and the penetrometer reading prior to dropping the mass. These values are labeled LP_max_ and LP_delta_, respectively, in this study. This process was repeated at three locations in each plot for each data collection event.

Divot resistance was measured by the TST. The device was zeroed out on the display before inserting the device into the ground for each data collection. After pulling the handle to shear off a piece of turf, the value of kgf on the display was recorded. Three such measurements were taken per plot in random locations in each of the three data collection events.

The GoingStick^®^ was used to obtain three measurements per plot in random locations in each of the three data collection events. This is distinct from the method used in most prior testing, where the only reported data are single values representing the average of three measures [34]. Penetration and shear from the GoingStick^®^ are labeled GS_P_ and GS_S_ in this study. Going Index was then calculated and is labeled GS_I_ in this study. The GoingStick^®^ was equipped with software version 2.29 with the “+33%” mode engaged. This software contains three modes: “jump” which was developed for jump racing, “flat” which was developed for flat racing in the UK, and “+33%” which was developed for flat racing in North America. The +33% mode is named as such because 33% firmer ground would be required to obtain the same reading as the flat mode in the GoingStick’s 1 to 15 scale.

To accurately determine gravimetric water content, one mixed sample was collected at each of the three data collection events from each plot with a 22 mm diameter sampling probe. The samples were mixed and then the composite sample was weighed on a balance and placed in a 60 °C oven for 16 h to remove moisture. Samples were weighed again upon removal from the oven. Gravimetric water content was calculated as the mass of water removed from the sample divided by the mass of dry soil.

### 2.4. Statistical Analysis

All statistical analysis was conducted using SAS system software, version 9.4 (SAS Institute, Inc., Cary, NC, USA). The proc corr function was used to generate Pearson’s correlation coefficients for each of the five simpler tools as compared to the four parameters measured by the OBST. The resulting correlation coefficients and associated *p*-values provide an indication of the strength and direction of a potential linear relationship between each set of measurements. The proc glm function was then used to generate linear models for each of the four measured OBST parameters. This was conducted for many different combinations of the five simpler tools as explained below. The simple tool variables listed in Table 3, Table 4, Table 5, Table 6 and Table 7 of Section 3 and Section 4 were included in each of the respective linear regression models. There were no covariates added to any of the models.

## 3. Results

There were 404 unique observations with each device described above. One OBST measurement was deemed erroneous and subsequently discarded along with the corresponding observations from the other tools. The mean and standard deviations for each parameter measured are presented in Appendix A (Table A2). The Pearson correlation coefficients (PCCs) between each of the simple tool measurements and the OBST parameters as well as the associated *p*-values are shown in Table 2 below.

A linear regression model was then generated for each OBST parameter that includes all the measurements from the simpler tools. The results are in Table 3, below.

**Table 3 animals-14-00038-t003:** Linear regression model for each of the four OBST parameters considering all measurements from simpler tools.

Simple Tool	Cushioning (R^2^ = 0.57)		Impact Firmness (R^2^ = 0.40)		Grip (R^2^ = 0.07)		Responsiveness (R^2^ = 0.07)	
	Estimate	*p*	Estimate	*p*	Estimate	*p*	Estimate	*p*
Volumetric Moisture Content	−0.056	<0.0001	0.517	<0.0001	0.079	0.039	−0.0001	0.647
Clegg Hammer (Average of Drops 2 and 3)	0.039	0.004	−0.181	0.131	0.021	0.702	−0.0009	0.027
Clegg Hammer (Drop 1)	−0.015	0.386	0.024	0.874	−0.031	0.654	0.0009	0.081
Longchamp Penetrometer Delta	−0.717	0.045	2.413	0.444	1.980	0.172	0.002	0.865
Longchamp Penetrometer Max	0.227	0.473	−0.541	0.846	−1.213	0.344	−0.0007	0.941
Turf Shear Tester	0.009	<0.0001	−0.067	0.001	−0.012	0.225	−0.0002	0.0006
GoingStick Penetration	0.004	<0.0001	−0.016	0.011	0.003	0.260	−0.0001	0.441
GoingStick Shear	−0.024	0.013	0.115	0.181	0.062	0.113	0.0003	0.323
Going Stick Index	0.00	-	0.00	-	0.00	-	0.00	-
Intercept	11.24	<0.0001	−77.51	<0.0001	0.13	0.970	0.64	<0.0001

To reduce the time demands on the maintenance personnel and enhance compliance, the objective is to identify the minimum data required to obtain an accurate representation of the surface conditions. The linear regression model was repeated to identify three simple tools which achieve the closest approximation of the OBST parameters (Table 4).

**Table 4 animals-14-00038-t004:** Linear regression model for each of the four OBST parameters considering three simpler tools which provide the most attractive R^2^ values.

Simple Tool	Cushioning (R^2^ = 0.51)		Impact Firmness (R^2^ = 0.39)		Grip (R^2^ = 0.07)		Responsiveness (R^2^ = 0.07)	
	Estimate	*p*	Estimate	*p*	Estimate	*p*	Estimate	*p*
Volumetric Moisture Content	−0.076	<0.0001	0.594	<0.0001	0.104	0.002	0.00007	0.794
Clegg Hammer (Average of Drops 2 and 3)	0.043	<0.0001	−0.219	<0.0001	−0.005	0.815	−0.0003	0.048
Turf Shear Tester	0.011	<0.0001	−0.069	0.0006	−0.006	0.503	−0.0002	0.001
Intercept	11.00	<0.0001	−75.49	<0.0001	4.45	0.110	0.64	0.0005

## 4. Discussion

While the simpler tools in this study have previously been used in other turfgrass applications, many have not been evaluated for equine surfaces. Previous studies have shown that tools developed for human athletes are insensitive to the higher loads in deep layers and the greater strain rates produced by a Thoroughbred at a gallop [35]. Comparing the simpler tools to the OBST on the plots used in this study provides a close approximation of the measurements on North American turfgrass racing surfaces. In particular, the use of synthetic reinforcing fibers has becoming increasingly common in order to handle heavier traffic due to the increasing popularity of turf racing. The plots at the Hancock Turfgrass Research Center used in this study represent a range of compositions. Some of these profiles are representative of current profiles with others have potential utility for future Thoroughbred tracks. With the increasing number of races run on turf these reinforced profiles may help maintain consistency.

The OBST measurements of the cushioning and impact firmness produced stronger PCC values and higher R^2^ values in the linear regression models than the grip and responsiveness. This finding was true for all the measurements from the simpler tools. Much of this can be attributed to the considerable noise in the grip and responsiveness measurements from the OBST. The noise is associated with the dynamic loading of the shoe and may be attributable to the complex physics of interface conditions. The frictional interfaces between a solid interface and granular materials exhibit stick–slip at the interface, which is sensitive both to small scale variation such as particle shape [36] as well as the dynamics of the interface such as the vibration of the load [37]. In an attempt to simplify the data analysis, bidirectional Butterworth filters were applied to the signals, as specified in ASTM F3400-19, which did not have a significant effect. The additional consideration of the dynamics of the interface would be beneficial but would need to consider both the dynamics of the machine and the properties and behavior of the surface. While the noise presents challenges associated with the data from the OBST, testing devices such as the TST or GS are less likely to represent the behavior of the racing surface because of the lower loading rate [7].

The VMC measurements displayed the strongest PCC values to cushioning, impact firmness, and grip, which indicates a strong linear correlation. The VMC also was the most significant contributor to the linear regression models for those variables as well. Racetrack maintenance personnel are also familiar with the VMC and many tracks already collect this data daily, as required by the HISA regulations. The importance of moisture measurements is evident. Relationships between the moisture content and hoof loads have previously been identified [28]. The strength of this correlation is such that moisture is the primary characteristic used for the characterization of Japanese racetrack conditions [17]. This observation is not limited to animal surface interactions but is also well-established in off-road vehicle mobility [38] where the loading and loading rates are similar to those in racetrack design.

The parameters measured by the OBST are biomechanically representative of the forelimb of a Thoroughbred at a gallop and so are also strongly influenced by the VMC. The cushioning of the surface is a shear failure in the top harrowed layer of the racetrack and is strongly dependent on the VMC [39]. Firmness is primarily determined by the layer under the harrowed surface, a partially saturated porous structure. The VMC determines if the pores are filled with air, water, or incompressible flow, which influences the response of the material, especially under dynamic loading [40]. Grip is also a shear-related phenomenon, not only in the granular material but also the frictional interface with the horse shoe. Frictional interfaces are sensitive to the effect of lubricants, with the well-established effect of water on the sliding between the grains of sand [41]. Thus, the VMC is the primary measurement to be taken on race days to characterize the surface and would be expected to impact the measurements made with the OBST.

The CIH value is calculated two different ways. The average of the second and third drops produced stronger PCC values as well as more significant contributions to the linear regression model than using the first drop alone. For this reason, collecting data in a similar fashion to ASTM F1936-07 is preferred (CIH_23_). Those data showed the second strongest PCC values for cushioning, impact firmness, and grip as well as the highest PCC value for responsiveness. The lightweight projectile and low drop height of the CIH results in a low impact velocity and low load. Since the strain rate sensitivity of partially saturated sand varies with the VMC, the ability to generalize results across a range of moisture content may be limited. The strain rate effects will differ in porous materials based on both the type of sand [42] and degree of saturation [43]. The averaging of the second and third drop does reduce the effect of the top layer of the material which, with the small mass of the projectile, can be heavily influenced by factors such as the grass cutting height and the presence of grass clippings, which would not be important to the performance of the surface when dynamically loaded by a 450 kg animal traveling at 15 m/s.

The LP data can also be calculated in two different ways. Reporting the difference between the maximum penetration value and the value prior to dropping the 1 kg mass (LP_delta_) resulted in a stronger correlation to the OBST parameters than the LP_max_. The LP uses a foot with a relatively large area to position the device which can result in a gap between the foot and the top of the soil. Unless the difference between the initial and final measurement is calculated, relatively unimportant factors like the grass cutting height would alter the initial measurement, which would be adjusted by using a differential measurement.

LP_delta_ and CIH_23_ displayed comparable correlations to the four OBST measurements, in particular, the cushioning and impact firmness. A key differentiator is that while the CIH has been used extensively in human sport applications, the LP has already been shown to be well-suited for race day measurements at turfgrass horse racing surfaces in New Zealand [18,29,30]. Furthermore, as these datasets were collected over a period of many years, they have shown the LP to be capable of assessing day-to-day variations in the racing surface, which is consistent with prior research [18,29,30]. Minimizing the spatial and temporal variation in a racing surface has already been shown to be key in the prevention of injury [5]. Thus, while the CIH and LP provide comparable results in the test boxes, the LP has already been accepted and has been shown to correlate to the performance and risk on active turfgrass horse racing surfaces. Unlike other measures of track conditions used in other jurisdictions, the New Zealand data are notable for being directly based on objective measurements [18], rather than using a subjective measure, which may include the interpretation of objective measures.

The TST had the highest PCC value for responsiveness among the five simple tools and was included in the three-device linear regression model, above, because of this relationship. However, as the R^2^ value for the linear model of responsiveness is never greater than 0.07, greater emphasis is placed on linear regression models which show stronger relationships such as cushioning and impact firmness. The TST is also heavier and more destructive, which also limits its potential usefulness for this application.

The GS, like the LP, was developed for the purpose of evaluating turfgrass horse racing surfaces. However, the device was less effective at approximating the parameters measured by the OBST based on the correlation and linear regression models. Of the 404 measurements collected with the GS, 53 were recorded at the upper limit of the shear value. This was the case even though the testing was primarily on cool weather grasses and the GS was set to the +33% mode, which was developed for the evaluation of North American surfaces. The range undoubtedly influenced the results and may indicate that while the GS may be useful in unreinforced soil, it may not be well-suited for the assessment of North American turfgrass horse racing surfaces, particularly if fiber or other reinforcement is present. These reinforcements are used to increase the shear strength of the surface without inhibiting drainage and are commonly found in North American turf tracks [24].

The GS is also more difficult to use than the LP and CIH since it is difficult to control the rate of loading and turf is strain rate-dependent [23]. While the LP and CIH measure the surface with a consistent energy input (falling mass dropped from a fixed height), the GS relies on the user to insert the tool to the full depth of the blade and pull back on the tool in the same manner every time. Different users, as expected, can then obtain different results with the GS on the same surface because of seemingly imperceptible differences in their rate of loading. As a result, it is difficult to compare data between racetracks to arrive at informed decisions about potential safety and performance implications. However, even when a single trained user took all the GS measurements in this study, the tools with a fixed energy input produced a closer approximation of the OBST parameters.

In addition to the distinctions from the fixed energy loading condition, the length scale over which the measurement is made differs between the LP, the CIH, and the GS. A turf track which is not damaged significantly but provides sufficient traction would have hoof prints which penetrate the surface but do not result in a divot. The penetration into the surface for the ideal surface would be the width of the shoe rather than the area of the hoof. The width of a racing plate is on the order of one centimeter. The depth of penetration would also be of the same order length scale. In this type of surface, the shoe would penetrate the surface to the depth of the frog but would not separate the turf in the area of the hoof during propulsion. The depth of the penetration and the probe on the LP has length scales on the order of a centimeter using a single drop. In contrast, the CIH projectile has a diameter of 50 mm with penetration dependent on the number of drops. The GS has a blade with dimensions of 100 mm long x 21 mm wide and is always pushed into a depth where the top plate is in contact with the surface. While the flat plate on the top of the GS is the approximate size and shape of a horseshoe, it is flat and the measurements are primarily influenced by the blade. In general, with a granular material, which has the characteristic lengths of grains that are on the order of 5 μm to less than 1 mm, the difference between the CIH and LP primarily becomes a concern when fibers with longer lengths are included. The depth of the CIH is, however, measuring a very different parameter since it is a repeated drop, so the length of measurement is dependent on the change in compaction, not the resistance to penetration, like the GS or LP. The penetration depth of the blade of the GS is greater than the other devices so the resulting measurement occurs at a different length scale.

Significant constraints related to time and labor availability limit racetracks’ ability to collect quality data on race days. The key aim of this paper is to ensure North American racetracks can collect sufficient data in a practical manner that can be used for evidence-based decision making. Surface condition measurements must be objective, repeatable, and efficient so as to be easily compared between surfaces [44]. As the data will be collected on active turf racing surfaces, minimizing the disruptions of the surface will also help to leave turf roots and thatch intact to support athletes during racing.

With moisture being identified as the primary simple measurement for the assessment of a racing surface, it would be useful to show comparisons to the OBST measurements. Moisture data can be collected quickly with virtually no disruption of the racing surface and produces a reasonable approximation of the cushioning and impact firmness. A linear regression model for the VMC is shown in Table 5 below.

**Table 5 animals-14-00038-t005:** Linear regression model for each of the four OBST parameters considering VMC only.

Simple Tool	Cushioning (R^2^ = 0.39)		Impact Firmness (R^2^ = 0.33)		Grip (R^2^ = 0.04)		Responsiveness (R^2^ = 0.02)	
	Estimate	*p*	Estimate	*p*	Estimate	*p*	Estimate	*p*
Volumetric Moisture Content	−0.126	<0.0001	0.859	<0.0001	0.114	<0.0001	0.0005	0.006
Intercept	17.61	<0.0001	−112.13	<0.0001	2.78	0.003	0.56	<0.0001

The LP measurements have already been shown to predict horse performance and injuries over an extended period [18,29,30]. The LP also has a minimal impact on the surface and, as a point measurement, it is well-suited to assess the temporal variations in the racing surface. A linear regression model for the VMC and LP_delta_ is shown in Table 6.

**Table 6 animals-14-00038-t006:** Linear regression model for each of the four OBST parameters considering VMC and LP_delta_.

Simple Tool	Cushioning (R^2^ = 0.45)		Impact Firmness (R^2^ = 0.34)		Grip (R^2^ = 0.05)		Responsiveness (R^2^ = 0.02)	
	Estimate	*p*	Estimate	*P*	Estimate	*p*	Estimate	*p*
Volumetric Moisture Content	−0.091	<0.0001	0.707	<0.0001	0.088	0.009	0.0004	0.148
Longchamp Penetrometer Delta	0.832	<0.0001	3.70	0.0009	0.626	0.200	0.005	0.208
Intercept	18.72	<0.0001	−117.09	<0.0001	1.95	0.085	0.55	<0.0001

To further improve the quality of the data, the CIH would be the next device to add to daily surface monitoring. A linear regression model for the VMC, LP_delta_, and CIH_23_ is shown in Table 7, below. The R^2^ values for these linear models indicate that the variation in the cushioning and impact firmness can be reasonably accounted for with these simpler tools. The grip and responsiveness, however, are not easily characterized by the simple tools used in this study. The OBST remains the primary device for assessing an active racing surface, and especially, the grip and responsiveness.

**Table 7 animals-14-00038-t007:** Linear regression model for each of the four OBST parameters considering VMC, LP_delta_, and CH_23_.

Simple Tool	Cushioning (R^2^ = 0.51)		Impact Firmness (R^2^ = 0.38)		Grip (R^2^ = 0.05)		Responsiveness (R^2^ = 0.04)	
	Estimate	*p*	Estimate	Pv	Estimate	*p*	Estimate	*p*
Volumetric Moisture Content	−0.063	<0.0001	0.552	<0.0001	0.088	0.018	0.00009	0.754
Longchamp Penetrometer Delta	−0.568	<0.0001	2.242	0.046	0.624	0.221	0.002	0.600
Clegg Hammer (Average of Drops 2 and 3)	0.041	<0.0001	−0.224	<0.0001	−0.0003	0.989	−0.0004	0.015
Intercept	13.89	<0.0001	−90.39	<0.0001	1.98	0.490	0.60	<0.0001

## 5. Conclusions

The volumetric moisture content is the one simple measurement which has the strongest correlation to the parameters measured by the OBST and is the priority for data collection on race days. The resulting model can be improved using either the Longchamp Penetrometer or the Clegg impact hammer. However, the Longchamp Penetrometer is preferred due to the previously established correlations to horse performance and injuries [18,29,30]. While further marginal refinements of the model are possible using CIH_23_ measurements, the use of additional tools increases the complexity and may be a practical barrier to adoption. The TST and GS do not appear to be well-suited to the representation of the OBST data.

While the OBST is currently used for the evaluation of equine surfaces on a seasonal basis, it appears that measurements from as few as two devices are a reasonable basis for daily decisions by racetracks and regulators on race days. These simple tools can be deployed on a large scale, based on the new federal regulations enabled by the HISA regulations [15]. Once the large-scale data collection has reached maturity, it can then be combined with additional data about the racetrack so as to continue additional information collection for large-scale epidemiological studies. Consistent objective data of a high quality has the potential to have a significant impact on our understanding of the racing surface’s contribution to the risk of injury and how racetrack personnel can make informed decisions on race days.

## Figures and Tables

**Table 1 animals-14-00038-t001:** Description of plots studied.

Name	Description
Well-graded sand	Sand profile with a broad distribution
Poorly graded sand	Sand profile with a narrow peak in the medium sand and fine sand size classes
7% silt and clay	Well-graded sand mixed with 7% silt and clay
9% silt and clay	Well-graded sand mixed with 9% silt and clay
15% silt and clay	Well-graded sand mixed with 15% silt and clay
9% silt and clay with Bermudagrass	Well-graded sand mixed with 9% silt and clay
Profile	75% well-graded sand mixed with 5% Canadian sphagnum peat and 20% Profile (ceramic particles made from illite clay and amorphous silica)
Zeopro	80% well-graded sand mixed with 10% Canadian sphagnum peat and 10% Zeopro (granules made from clinoptilolite and synthetic apatites)
Turfgrids	Well-graded sand mixed with randomly oriented fibrillated polypropylene fibers
Ventway	80% well-graded sand mixed with 20% Ventway (randomly oriented cylindrically shaped rubber particles)
StrathAyr	StrathAyr specified root zone mixed with polypropylene fibers
Grassmaster	Well-graded sand with polypropylene fibers sewn vertically into the established turf. Fibers inserted 2 cm on center to a depth of 20 cm
Hummer Turftiles	Reinforced sod with shredded nylon carpet fibers to a 5.1 cm depth established on top of well-graded sand
Motzgrass	Reinforced sod with polypropylene fibers sewn into a backing established over a Motz-specified rootzone
Sportgrass	Reinforced sod with polypropylene fibers sewn into a synthetic backing established on top of well-graded sand

**Table 2 animals-14-00038-t002:** Pearson correlation coefficients (PCCs) for comparing each simple tool to the four OBST parameters.

Simple Tool	Cushioning		Impact Firmness		Grip		Responsiveness	
	PCC	*p*	PCC	*p*	PCC	*p*	PCC	*p*
Volumetric Moisture Content	−0.63	<0.0001	0.57	<0.0001	0.21	<0.0001	0.14	0.006
Clegg Hammer (Average of Drops 2 and 3)	0.62	<0.0001	−0.51	<0.0001	−0.14	0.005	−0.19	0.0002
Clegg Hammer (Drop 1)	0.53	<0.0001	−0.44	<0.0001	−0.12	0.012	−0.14	0.004
Longchamp Penetrometer Delta	−0.56	<0.0001	0.45	<0.0001	0.17	0.003	0.13	0.013
Longchamp Penetrometer Max	−0.53	<0.0001	0.43	<0.0001	0.15	0.0005	0.12	0.008
Turf Shear Tester	0.37	<0.0001	−0.32	<0.0001	−0.09	0.080	−0.22	<0.0001
GoingStick Penetration	0.41	<0.0001	−0.28	<0.0001	0.03	0.496	−0.11	0.027
GoingStick Shear	−0.04	0.383	0.05	0.278	0.12	0.013	0.002	0.959
Going Stick Index	0.28	<0.0001	0.18	0.0004	0.08	0.091	−0.08	0.120

## Data Availability

Data are contained within the article.

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
