# Peer review of "A Comparison of Devices for Race Day Characterization of North American Turfgrass Thoroughbred Racing Surfaces"

_animals, 2023, doi:10.3390/ani14010038_

Round 1

Reviewer 1 Report

Comments and Suggestions for Authors

1.       A brief summary outlining the aim of the paper, its main contributions and strengths

The paper aims to investigate the use of simple testing devices to measure racetracks on a daily basis with the aim of providing a practical assessment method that provides high quality data without the use of the OBST which is time consuming, costly and results in significant damage to the track.

The paper describes an experimental project that investigates the use of equipment in an experimental setting. The results allow the authors to recommend the use of three measurements that correlated well to the OBST and may have merit within the industry, as methods to help practitioners manage and maintain racetracks.

2.       General concept comments

Abstract

Please refer to comments within the rest of the paper and ensure any modifications are supported in the abstract. Specific comments are presented below for this part of the paper.

Introduction

The introduction is rather disjointed, and some elements appear repetitive (see specific comments below). The mention of sports turf tools seems to almost be an afterthought. There is a need for some evaluation of the literature and gaps in knowledge of these tools. The introduction is rather light on content but starts with an emphasis on injury and safety but this is not what the paper focuses on. The first 1.5 paragraphs should be reworked to place more emphasis on what the paper is about (methods of measuring racetracks). It would benefit from some discussion of the literature on the effect of rootzones as this study uses 15 different rootzones.

Methods

The methods need some work to improve clarity and avoid repetition. There seems to be some repetition between section 2.2 and 2.3. Whilst reading section 2.2 the reader finds themselves with multiple questions that are subsequently answered in section 2.3 but with better structure there could be better clarity. The tools are all described in section 2.2 but some information is missed and further information is provided in the next section (2.3). These two sections need to be reworked to provide a better logical order and narrative.

There are some sections that are not well considered whilst others could follow a more succinct order and could avoid significant levels of repetition. The section for the statistical analysis requires some significant improvement. There are several changes in tense throughout.

Results

Table titles require additional information to allow the reader to fully understand what is being presented. Please explain abbreviations in the title or include a short glossary below each table; as it stands, it makes it hard for the reader to interpret what was found.

The text for the results needs to be better connected to what the tables are presenting, thus highlighting key findings, which is really lacking here. Please revisit this section and see specific comments line by line.

The results would benefit from a table that presents something like mean and variance that may pull out some of the nuances of the data presented here. Looking at the various rootzones / soil profiles, it would be possible that for some of these pieces of equipment, there would be quite a bit of variation from data collected from these. The reader would benefit from seeing this information and it may be that the regression models could be presented in a slightly different format to accommodate another table.

Discussion needs significant work and lacks critical evaluation and comes across as rather descriptive. There is a brief presentation of each piece of equipment with limited evaluation nor synthesis, more just explaining the results. There is a need for a more scientific discussion section that refers more thoroughly to previous literature. At this stage the discussion is not suitable. There seems to be limited consideration for understanding what each tool is doing and what OBST parameters it should relate to. Please see comments line for line to help you develop a much stronger discussion that is better supported by the literature and demonstrates a focused critical evaluation of the equipment and the findings.

Conclusion – The claims made in this conclusion are not supported from the paper. The conclusion refers to injury which is not included in this study. The conclusion is not concluding the paper. Please revisit this and see specific comments that have been made.

3.       Specific comments referring to line numbers, tables or figures that point out inaccuracies or lack of clarity

Abstract

Line 16-19 not sure these tool approximate to equine biomechanics (horse biomechanics is not discussed in the paper in relation to the simple tools)

Introduction

Line 47-57 evidence for these statements is limited, needs better support from the literature

Line 58-60 Why daily basis? Needs to be explained or supported

Line 60-62 It could be helpful if there is a brief explanation as to what these requirements are and why

Line 68-69 OBST is referred to but it would benefit from being better connected to what has bee said in the previous paragraph

Line 69-74 Is this for dirt/synthetic and turf in N America? It needs to be clearer as to whether it is used for all types of racing prior to the race and is this every race. Rather vague.

Line 76 The goal of who? The study?

Line 80-81 and Line 84-85 is repeating the same information, please rework this to avoid repetition

Methods

Line 99 included not include

Line 99-108 The test configurations need to be explained and justified as to why these plots were used. There is a point made that they were already set up from previous studies, but are these the types of surfaces that a racetrack would be laid on or not? This information is integral to the study and needs explaining.

Line 99-100 this sentence is very vague, if these fibre reinforcements have been in since the 1990s they will likely have degraded but equally, the original work that is referred to, was set up as sports turf plots for athletic fields. Did all plots have the same ratio of Kentucky Bluegrass and Bermudagrass?

Line 104-105 please explain what seasonal maintenance is.

Line 112-113 parameters were calculated – but were calculated from what? Accelerometer, load cell, potentiometer? This would be useful here.

Line 118 FieldScout: resolution and electrical conductivity?

Line 142-143 why at a depth of 40 mm? It was fitted with a 50 mm shearing plate – was this unique to this study or normal practice with this tool?

Line 140 - Longchamp penetrometer not directly correlated with ms injuries!? Just that firmer tracks were associated with greater risk and care should be taken as this paper did not discuss the Lonchamp penetrometer and just reported data according to ranges. I could not access the papers from 1996 to confirm these findings but please could you check that these are reported correctly.

Line 147 in reference to work reported using the TST, ‘those surfaces’ – could you define what types of surfaces.

Line 154 Can you explain what SI units the GS is converted to? Is it mm and N?

Line 154-156 Can you explain the scale of the Going Index? Some reference to this would be beneficial here as this is a piece of equipment that is relatively unique and probably not well known to the reader. A more comprehensive explanation would help – perhaps refer to Mumford?

Line 158-165 can you give a brief explanation for 1 hour and 4 hours? There has been no clear mention of this until now.

Line 168 random locations – were they decided in advance? This applies to all the other times that this is explained. If the methods are restructured, it may be worth considering how repetitions such as this can be avoided

Line 174-184 Can you explain what is being assed with the 1st drop versus the average 2nd and 3rd drop?

Line 199 How did you know if it was a 45o ­angle? Training? One researcher to conduct all tests?

Line 202 please can you explain what +33% means

Line 206 this is not in metric but everything else is?

Line 205 was this one sample per plot per collection event?

Line 211-218 Statistical analysis needs some work. For the linear models, can you state the factors that were used and identify if there were any covariates added? There is a need to be more explicit about what has been done and why for each of the regression models that were run. The ones presented in the discussion would be better placed in the results section and use of statistical analysis section in methods to explain what was done and why.

Line 217-218 – extremely vague, please could this be explained in more details.

Results

Line 232-234 is explaining what was carried out in the results section. It would help having some of this explanation in the statistical analysis section to demonstrate how the statistical analysis was conducted. Vague to suggest that the ones that demonstrated the closest approximation.

Line 323 – 340 Can the regression models be presented in the results section rather than being presented halfway through the discussion? The discussion becomes descriptive by presenting further results and feels as though it comes as a surprise. It would be a good idea to reconsider the structure of the methods and results to better accommodate this.

Line 234 The linear regression was repeated to identify three simple tools - spelling

Line 240-251 this would be better placed in the introduction and would help the reader understand why these types of rootzones have been used. There is nothing of this in the introduction which would be a better place to put it. No evidence to support the use of synthetic fibres in racing? There is a need for evidence to support this statement.

Line 252 – 254 do you mean the PCC between C and IMF against the simpler tools, this is not clear, please reword

Line 254-257 – noise in the OBST but also the simpler equipment may not be measuring responsiveness and perhaps there should be a discussion about this in relation to the GS and TST as these were both measures of shear resistance that are influenced by grip. Are you suggesting that the readings from the OBST for grip and responsiveness account for a weak PCC?

Line 260-262 is supporting the result of VMC by stating that personnel are familiar with this but this is not a reason for the high PCC. Please revisit this paragraph as the sentences do not support each other. Can you explain the reason for why VMC is relevant to cushioning and impact firmness? In line 262-263 you state that the importance of moisture is evident but there is no evaluation or discussion of this. Can you suggest why grip may not show a strong PCC? As a reader I would appreciate seeing the actual data and the variance in the data. Using 15 different types of rootzone should be explored when discussing grip but without seeing this data it is not possible to ascertain other factors that may be going on here.

Line 263-264 is not supported and needs to be considered in more depth.

Line 266-271 just describes the findings but does not address the differences in CIH1 and CIH23, nor the weight of the CIH and how this might influence the results, please address this. You state that the PCC is highest for grip and responsiveness but just because it is the highest does not mean it is relevant (a PCC of 0.1 is a weak correlation, why would this be relevant? What does the CIH measure and how will a vertical drop hammer using an accelerometer tell you about grip and responsiveness? This needs to be explained and discuss in relation to the literature (there is quite a bit out there on turf surfaces using a CIH).

Line 272-274 his is just describing the results. The difference between LP max and delta need explaining – delta means you account for the value prior to dropping and in an ideal situation the LP will be reading zero prior to dropping it but on undulating surfaces this is not always possible. This should ideally have been explained in the methods (Line 133-131) so it is clear to the reader before they get to the results.

Line 277-280 – please refer to the literature that explains this, the research you refer to does not look at this in detail. You state that data was collected over many years but there is a need for this to be better explained. The LP is used for NZ racetracks as one measure to assist in assessing racetrack going but the LP itself has not been specifically shown to be capable of assessing day to day variation in the paper [21] not yet sure about the papers [19,20]. Please check this.

Line 282-284 reference [21] does not show that LP correlates to injury (as mentioned previously) so this needs to be revisited and rewritten to provide a more accurate commentary.

Line 285-290 A correlation of 0.22 would normally be considered weak / negligible… why would the LP be relevant to responsiveness then this needs to be discussed in light of previous work in sports turf.  There is a need to evaluate this tool here and justify using or not using it.

Line 296 – there was no explanation as to what the +33% means and if this is presented here it needs to have been explained earlier and then referred back to here.

Line 291-300 can this be discussed in light of previous work? Mumford for instance?

Line 308-310 Generalising to all tools here- can you talk about variation seen in user on the plots so it provides the reader with some context and detail that could be useful in the future? Compare to use and research in sports turf. Please discuss in more detail.

Line 318 repeated earlier

Line 330-337 can you discuss the relevance of these functional properties to the surface (for maintenance) and the horse? There is a need for some discussion as to what these measurements tell you about the surface for the horse as this is missing and should be discussed in view of the results from the regression models.

Line 336-337 stating more research is needed warrants a discussion, please explain what that more research is or avoid stating it

Conclusion

Line 342 -343 repeated three times in the paper but not what is known from the reference [21] about injuries (see comments earlier on).

Line 344-345 not fully discussed how VMC is complemented by LP? Please revisit this and make sure you discuss this more fully should you wish to keep this in the conclusion

Line 349 A stronger discussion would help justify this recommendation but from what has been presented, this conclusive statement is not fully supported.

Comments on the Quality of English Language

N/A

Reviewer 2 Report

Comments and Suggestions for Authors

1)      A brief summary (one short paragraph) outlining the aim of the paper, its main contributions and strengths.

Assessment of the physical properties and condition of turfgrass (at equestrian events /races) is essential to minimise injury potential and optimise performance.  There is however a continuing discussion about the appropriate assessment methodology(s).  Currently there are a range of equipment which assess individual ground reaction forces (GRFs) but few (if any) are considered to truly reflect the forces that are observed in the horses limb. The discussion is further confounded by the need for equipment to also be suitably portable and ‘simple’ for reliable use by industry practitioners. The ability of these pieces of equipment to reliably reflect these GRFs has not been sufficiently established and could be feeding inaccurate data to the industry and creating an unknown safety risk to horse and rider.

This paper directly addresses this issue by testing a range of the widely used equipment against the accepted ‘gold standard’ tool (the OBST) for assessing turf GRFs.  It is a necessary step in establishing the correct protocols for assessing turfgrass in a range of equestrian sports.

2) General concept comments.

Article: highlighting areas of weakness, the testability of the hypothesis, methodological inaccuracies, missing controls, etc.

The introduction generates a compelling discussion about the current state of the industry and the need for collecting the appropriate data at the pre-meeting stage.  My only comment is there is no overview or specific mention of the tools that are being focused on in the study or much background as to why these particular tools have been chosen. There is a good breakdown of each piece of equipment in the methods section, but detail on why these tools are either not accepted or have been accepted by the industry is not apparent. The GoingStick in particular (which has been purposely designed with the racing industry in mind) is also a notable omission from the overview.    

The methodology is generally good – the choice of equipment is comprehensive and broadly covers the main range of tools which are currently available to test a surface. It is always difficult to get absolute consistency in outdoor trials – climatic variations can often be a confounding factor.  This methods seems to have accounted for them effectively.  The GoingStick and the TFT are both prone to operator differences; did the same user operate the equipment each time ?  If not, it may be worth factoring this into any analysis you are undertaking.

The Statistical Analysis section is a little brief – what were the factors used in the linear models, were there any covariates added? A fuller description of the variables / factors entered into each model would be a useful addition.  

The results are clear and contain a sufficient level of information to demonstrate the major relationships.  I think the propensity for acronyms hampers the clarity in places (particularly the results tables) – please consider reverting to the full titles within the tables. 

The opening paragraph sets the context to the findings effectively although it especially highlights the N. American context.  These results have a broader global implication and some effort to emphasise this would be useful and also highlights the importance/usefulness of these findings.

Lines 252 to 271 are primarily descriptive and sets the scene but does not evaluate the results in an appreciable depth.  The discussion does then start to evaluate more effectively.  The section on the GoingStick (lines 291-300) is particularly useful but it requires a little more context. The writers highlight that there are potential situations where the GS may not give accurate results – this needs  more context – what/where might this be an issue? Smaller racecourses? Eventing tracks?

The addition of more regression models in the discussion is a slightly controversial way of displaying them. I understand the logic behind this approach but these results need to be moved to the results section and the discussion rearranged to accommodate it. I think it wise to marginally strengthen the introduction to accomodate the inclusion of this element.  

3) Specific comments referring to line numbers, tables or figures that point out inaccuracies within the text or sentences that are unclear.

These comments should also focus on the scientific content and not on spelling, formatting or English language problems, as these will be addressed at a later stage by our internal staff.

Line 106 – reference missing

Line 113 – isn’t cushioning a factor that combines moisture and peak vertical load?

Line 224 – reference link does not work

Line 228 – reference link does not work

Line 298 – expand this point about unreinforced soils and add some context – in what settings would you find these soils?

Line 305-310 – great point, some mention of this in the introduction would be a good link

Line 323-340 – move the results into the results section and re-arrange discussion to accommodate.

Line 386/387 – reference link does not work

Reviewer 3 Report

Comments and Suggestions for Authors

This manuscript compares methods for determining the condition of tracks used for thoroughbred horse racing. Determining best practise for track assessment is important given the relationship between track condition and various veterinary conditions in racing horses (including but not limited to catastrophic injury) that have been previously identified and for compliance with the emerging HISA conditions applicable to racing in the USA. Assessments of track conditions are made throughout the world so the outcomes of this study are potentially of international relevance.

Introduction

The introduction provides an excellent overview of the importance of the topic. It provides a sound explanation of the basis for the study, namely the need to establish the suitability of simple-to-use tools compared with the gold standard.

It would be useful to define ASTM 3400-19 and to provide some background on how it relates to the relevance of the study.

Elaboration on the sentence “The five simple commonly used tools from athletic turf applications…” would be helpful. Presumably the authors are referring to tools used in the assessment of turf race tracks? If so, it may be simpler to say that they are tools used to assess to a thoroughbred tracks, and if that is not the case then a better explanation would be helpful. I think this is relevant given the statements in the first paragraph of the discussion. What was the process by which these tools were selected? How was common determined - was this on the basis of a survey or is it from previously published work? Given that the ASTM is an international standard, can the authors state whether these tools are commonly used in the USA or more widely used internationally?

Methods

The methods provide a good explanation of the plots and the tools used (and the specific features of each). The information provided is certainly sufficient to allow replication of the tools used and is probably sufficient to enable replication of the plots. Presumably the nature of the turf grown in each of the different plots could vary from place to place, but this is an unavoidable challenge to the ability to replicate the study should someone desire to do so in the future. It in no way detracts from the quality of the study or its findings.

The experimental procedure is well designed and well described and the statistical analysis is appropriate.

Results

The results report appropriately the findings resulting from the analysis of the collected data.

Discussion

It is debatable whether some of the content of the discussion should appear in the results. Others may have a different view but I feel that the presentation of the results in the discussion is appropriate as it makes it easier for the reader to make sense of the narrative that is unfolding.

The discussion provides useful advice on the practical application of the findings of this study. It is appropriate to the aims of the study and to the data collected. The conclusion is sensible and clearly shows how the findings of the study relate to the aims stated in the introduction.

General

I think that this is a very useful manuscript. The design of the study is simple but practical and enables informed conclusions to be drawn. The quality of the writing is good both in terms of meeting requirements to enable replication and in clearly describing the rationale and outcomes of the study.Although I have rated interest to readers as low that simply reflects my view that this is a very specialised field. I believe that people in that field will find the information very useful. 

Round 2

Reviewer 1 Report

Comments and Suggestions for Authors

Thank you for addressing comments and your responses. There are a few minor points included below that deserve some attention prior to publication.

Specific comments referring to line numbers in V2 of the manuscript:

Line 246 You have provided me with an explanation of what +33% means but my question was aimed at the paper rather than for me personally. Can you provide a brief explanation here for the reader so that it is clear to them?

Line 250-254 The response related to whether one or more samples were collected per plot for gravimetric water content was given to me as the reviewer but can this be included in the methods section (i.e. three samples were collected from each plot).

Line 301-306 (for the comment about noise from the OBST). The response that has been given is okay but does not fully address my comment. It is fair enough that there is noise but is it possible that you can provide a brief discussion as to why certain pieces of equipment would be less suitable for measure say grip or responsiveness? Can this be discussed briefly here in relation to the literature?

Line 313-317 I fully agree that it is well established that VMC is an important simple measure but it would be expected that there is some explanation here as to why this is so relevant and important to cushioning, impact firmness and grip. Currently it is stated that other papers have found this too but can you explain why (this may be a good point to refer to how when you consider VMC you should also know something about soil type / particle size difference and that this could be investigated further when more data has been collected - see comment about the conclusion).

Line 324-327 does not quite make sense, please revisit. There is some useful information added here but just check the sentence.

Line 333-339 You state that LP can be calculated in two different ways but you have not explained both of these methods and it may be worth adding. You state that LPdelta is stronger, but do not explain what it is stronger than, can this be added here? You have explained the difference between first and final measurement for LP but is there a strong correlation between all OBST parameters and what does a penetrometer equate most to (i.e. cushioning, impact firmness, grip or responsiveness)… i.e. what does this mean and why?

Line 364 Can this explanation be included in the manuscript rather than just being given to me as the reviewer?

Line 390 Can this be in mm as in the methods and also comparable to the other measurements given in this paragraph?

Line 400-401 Is there a word missing as the sentence seems incomplete?

Line 445 'as' is missing - such as particle size difference.. 

Line 433-448 Conclusion should only conclude what has already been discussed and within the discussion there is no mention of particle size distribution and race times which would be expected in view of it being important enough to be a concluding comment. Can this be addressed? VMC has been identified as being the most important simple test but there has not been any mention of how soil type and particle size distribution will influence this. Perhaps this can be added within the discussion at around Line 307-317 which would provide opportunity to explain how in future, this could be examined further. Some explanation as to why race times are relevant would be helpful within the discussion if it is to be included here in the conclusion.
